# Transcriptomic Analyses of *Camellia oleifera* ‘Huaxin’ Leaf Reveal Candidate Genes Related to Long-Term Cold Stress

**DOI:** 10.3390/ijms21030846

**Published:** 2020-01-28

**Authors:** Lingli Wu, Jian’an Li, Ze Li, Fanhang Zhang, Xiaofeng Tan

**Affiliations:** 1Key Laboratory of Cultivation and Protection for Non-wood Forest Trees of Ministry of Education and the Key Laboratory of Non-Wood Forest Products of Forestry Ministry, Central South University of Forestry and Technology, Changsha 410004, China; wulingli0307@163.com (L.W.); lize1853@163.com (Z.L.); ZhangFH1013@163.com (F.Z.); 2Engineering Technology Research Center of Southern Hilly and Mountainous Ecological Non-Wood Forest Industry of Hunan Province, Changsha 410004, China

**Keywords:** *Camellia oleifera*, long-term cold stress, transcriptome sequence, DEGs

## Abstract

‘Huaxin’ is a new high-yielding timber cultivar of *Camellia oleifera* of high economic value, and has been widely cultivated in the red soil hilly region of Hunan Province of the People´s Republic of China in recent years. However, its quality and production are severely affected by low temperatures during flowering. To find genes related to cold tolerance and further explore new candidategenes for chilling-tolerance, Illumina NGS (Next Generation Sequencing) technology was used to perform transcriptomic analyses of *C*. *oleifera ‘*Huaxin’ leaves under long-term cold stress. Nine cDNA libraries were sequenced, and 58.31 Gb high-quality clean reads were obtained with an average of 5.92 Gb reads for each sample. A total of 191,150 transcripts were obtained after assembly. Among them, 100,703 unigenes were generated, and 44,610 unigenes were annotated. In total, 1564 differentially expressed genes (DEGs) were identified both in the A_B and A_C gene sets. In the current study, Gene Ontology (GO) and Kyoto Encyclopedia of Genes and Genomes (KEGG) pathway enrichment analyses were performed, andrevealed a group of cold-responsive genes related to hormone regulation, photosynthesis, membrane systems, and osmoregulation; these genes encoded many key proteins in plant biological processes, such as serine/threonine-protein kinase (STPK), transcription factors (TFs), fatty acid desaturase (FAD), lipid-transfer proteins (LTPs), soluble sugars synthetases, and flavonoid biosynthetic enzymes. Some physiological indicators of *C*. *oleifera* ‘Huaxin’ were determined under three temperature conditions, and the results were consistent with the molecular sequencing. In addition, the expression levels of 12 DEGs were verified using quantitative real-time polymerase chain reaction (qRT-PCR). In summary, the results of DEGs analysis together with qRT-PCR tests contribute to the understanding of cold tolerance and further exploring new candidate genes for chilling-tolerance in molecular breeding programs of *C. oleifera* ‘Huaxin’.

## 1. Introduction

*Camellia oleifera* Abel. is a member of the genus *Camellia* in the family Theaceae. In China, cultivated *C*. *oleifera* is one of the four major woody oil species, followed by the tung tree (*Vernicia fordii* Hemsley), walnut (*Juglansregia*), and Chinese tallow tree (*Sapium sebiferum*) [1,2]. The tea oil extracted from *C. oleifera* seeds is an edible oil known as “eastern olive oil,” because of its high nutritional value and health benefits [3]. Tea oil has a similar chemical composition to that of olive oil; both contain high amounts of unsaturated fatty acids [4]. As an evergreen broadleaf shrub or small tree, *C*. *oleifera* is widely distributed in the subtropical mountainous areas of the Yangtze River basin and South China [5]. The Jiangxi, Hunan, and Guangxi provinces are the main habitat areas, accounting for 76% of the total area of production in the country [6]. With the rapid development of the *C*. *oleifera* industry, large areas of red soil hilly region of southern China have been planted with *C*. *oleifera* in recent years.

Hunan has a subtropical monsoon humid climate with four distinct seasons, where the mean air temperature in autumn and winter is 10–12 °C, and the mean January air temperature is 4–8 °C. *C*. *oleifera* blossoms at daily average temperatures ranging from 10 to 20 °C, and 8–14 °C is the optimum temperature range for flowering [7]. Unlike most of the woody plants in China, which usually bloom in spring (from March to May in the same year), *C*. *oleifera* usually flowers from November of that year to January of the next year. Hence, damage due to low temperature stress at the flowering phase has severely hindered oil production of *C*. *oleifera* in the hilly red soil region of southern China. 

Temperature, a very important ecological factor for plant growth, affects the distribution, productivity, and survivability of plants [8]. Previous studies have confirmed that the expansion of tea plant (*Camellia sinensis*) cultivation has been restricted due to temperature, and cold stress might cause the fall of flowers and fruits, leading to a reduced yield [9,10]. Furthermore, physiological and phenotypic changes occur in response to low-temperature stress, such as decreased chlorophyll content, photosynthetic rate, and electron transport rate (ETR) [11]. Tropical or subtropical plants subjected to air temperatures less than 10 °C are easily affected by chilling stress [12]. Cold and freezing weather would cause a large number of flower and fruit to fall, and it is also not conducive to pollination and young fruit formation because pollen cracking is inhibited when the temperature drops below 8 °C [7]. Woody plants in temperate climates achieve their cold tolerance ability to survive cold stress through a series of processes induced by seasonal acclimation to low temperatures, known as cold acclimation [13]. Cold acclimation triggers various adaptive molecular mechanisms in response to long-term cold stress [14], such as protection and stabilization of cellular membranes, enhancement of antioxidant mechanisms, synthesis and accumulation of osmotic protective substances, and unique cryoprotective proteins [15]. There is a positive correlation between the accumulated temperature at the flowering stage and the yield of *C*. *oleifera* [16]. 

In 1977, a *C. oleifera* research team found a wild tea-oil tree with good fruit performance and strong resistance to disease in Chaling County, Hunan Province. From 1978 to 2009, through comparison and regional testing of 84 *C. oleifera* clones with good characters, the clones with the best comprehensive characters were screened out, and named ‘Huaxin’ by the forest variety Committee of the State Forestry Administration. ‘Huaxin’ is a new *C*. *oleifera* cultivar, which has high and stable yields, strong disease-resistant ability, and precocity [17]. Because of its high yield and good economic returns, it has been widely cultivated in the red soil hilly region of Hunan Province in recent years. However, *C*. *oleifera* usually flowers in autumn and winter, and it is easily affected by cold temperatures during flowering [18]. Previous studies showed that low temperature inhibited the normal opening of ‘Huaxin’ flower buds, and an appropriate temperature was favorable for the growth of young fruits [19]. To improve its cold resistance and extend its cultivation to colder areas in the north, it is necessary to identify the candidate genes related to long-term cold stress and understand the regulation of gene expression of *C*. *oleifera* ‘Huaxin’ during flowering. This improved understanding also provides a basis for cultivar improvement and molecular breeding of *C*. *oleifera*. 

Approximately 120 species belong to the genus *Camellia*, and *C*. *oleifera* and *C*. *sinensis* are the two most representative species, with high economic value [5]. However, the completely sequenced genome for these species remains unavailable. For non-model plant species that lack gene information, transcriptome sequencing is an alternative rapid approach for studying the molecular mechanisms of physiological processes. Although several transcriptomic analyses in tea plants (*C*. *sinensis*) have been reported [20,21,22], a few transcriptomic studies of *C*. *oleifera* have been undertaken. In one study, the transcriptome of *C*. *oleifera* ‘Huaxin’ was sequenced and de novo assembled using different plant tissues, which revealed some candidate genes associated with lipid metabolism [23]. Though atranscriptome profile of wild oil-tea camellia from various latitudes and elevations under low environmental temperatures in winter has been reported [5], transcriptomic studies of cultivated *C. oleifera* ‘Huaxin’ under controllable temperature conditions are few at present. 

A previous study has shown that low temperature had a great influence on the phenotype and physiology of *C. oleifera* ‘Huaxin’ leaves [19]. To explore the mechanism of *C*. *oleifera* ‘Huaxin’ leaves in response to long-term cold stress, a fully characterized *C*. *oleifera* ‘Huaxin’ transcriptome using Illumina NGS technology combining with physiological experiments was performed. This led to the discovery of several differentially expressed genes (DEGs) in response to cold and, ultimately, to a better understanding of the molecular mechanisms involved in the *C. oleifera* ‘Huaxin’ adaptation to low temperatures. Some candidate genes related to cold acclimation were identified that provide a basis for cultivar improvement and further molecular breeding of *C. oleifera*.

## 2. Results

### 2.1. Morphological Observation and Physiological Indexes of C. oleifera ‘Huaxin’ Under Three Temperatures

There were substantial differences in leaf color among three temperature treatments. The leaves of *C*. *oleifera* ‘Huaxin’ were thick and dark green under treatment A, but turned yellow at the edges under treatment B, and were slightly pale under treatment C (Figure 1a–c). In addition, the chlorophyll contents (Chl), net photosynthetic rate (*P_n_*), and electron transfer rate (ETR) were significantly different under three treatments; all values were the lowest for treatment B, followed by treatment C, and the value of treatment A was the highest (Figure 1d–f). However, the soluble sugar contents of *C*. *oleifera* ‘Huaxin’ under treatments B and C were significantly higher compared with treatment A, at 41.67% and 33.33% higher, respectively (Figure 1g). The starch content of *C*. *oleifera* ‘Huaxin’ under treatment A was the highest among the treatments, at 85.67% and 24.73% higher than treatments B and C, respectively (Figure 1h). The relative water content (RWC) of treatment B was the lowest. Compared to treatment A, RWC of treatment B was significantly decreased by 11.69%; however, no significant differences were found between treatments A and C (Figure 1i). The galactose content of *C*. *oleifera* ‘Huaxin’ under treatment B was the highest among the treatments, at 42.56% and 26.36% higher than treatments A and C, respectively (Appendix A).

### 2.2. Transcriptome Sequencing and Assembly

An overview of the RNA-Seq reads derived from nine cDNA libraries (A1, A2, A3, B1, B2, B3, C1, C2, and C3) are shown in Additional File 1: Appendix A. After quality control, 58.31 Gb high-quality clean reads were obtained with an average of 5.92 Gb reads for each sample, and the percentage of Q30 base in each sample was >92.56%. The proportion of mapped reads per library ranged from 70.43% to 74.59%. A total of 191,150 transcripts were obtained from the clean reads with an N50 length of 1690 bp and a mean length of 1034.37 bp. Among them, 100,703 unigenes were generated, and the average length of the unigenes was 790.43 bp. There were 64,762 (64.31%) unigenes with a length >300 bp, while the lengths of 23,846 (23.68%) unigenes were >1000 bp. The length distributions of the transcripts and unigenes are listed in Table 1. The assembly quality of the transcriptome was rated as satisfactory.

### 2.3. Gene Annotation and Functional Classification

BLAST was used for unigene function annotation with common functional databases such as NCBI non-redundant protein sequences (NR), a manually annotated and reviewed protein sequence database (Swiss-prot), Gene Ontology (GO), Clusters of Genes (COG), EuKaryotic Orthologous Groups (KOG), the Kyoto Encyclopedia of Genes and Genomes (KEGG), eggNOG, and Protein family (Pfam). Out of the 100,703, 44,610 (44.30%) were successfully matched to homologous sequences in at least one of the databases mentioned above. Among them, 13,161 (29.50%), 28,213 (63.24%), 14,824 (33.23%), 23,416 (52.49%), 26,264 (58.87%), 24,203 (54.25%), 40,098 (89.89%), and 43,518 (97.55%) annotated unigenes were obtained from the COG, GO, KEGG, KOG, Pfam, Swiss-Prot, eggnog, and NR databases, respectively (Table 2).

The NR database produced the largest number of annotations with a total of 43,518 unigenes. Based on the results of the smallest Blast E value, *C*. *oleifera* ‘Huaxin’ showed the most matches to *Vitis vinifera* (12.14%), followed by *Quercus suber* (4.71%), and *Juglans regia* (2.36%) (Figure 2a). The GO is an international standardized classification system of gene functions. It provides a dynamically updated standard vocabulary to fully describe the possible molecular functions of gene products, the cellular environment, and the biological processes. A total of 28,213 unigenes were annotated and classified into 58 GO pathways (Figure 2b) in three categories involving Cell Component (CC), Molecular Function (MF), and Biological Processes (BP). The top three GO terms for the CC category were ‘cell’ (44.78%), ‘cell part’ (44.76%), and ‘membrane’ (36.49%); for the MF category ‘catalytic activity’ (51.81%) ranked the first, followed by ‘binding’ (47.57%) and ‘transporter activity’ (7.43%); for the BP category, the top three GO terms were ‘metabolic process’ (52.54%), ‘cellular process’ (48.50%) and ‘single-organism process’ (33.13%). In addition, genes were also highly represented in three GO terms including ‘organelle,’ ‘response to stimulus’, and ‘biological regulation’. The KOG classifications were based on genetic orthologous and evolutionary relationships. In total, 23,416 unigenes were grouped into 25 functional classifications based on the KOG database (Figure 2c). Among these classifications, the ‘general function prediction only’ (26.18%) ranked highest, followed by ‘posttranslational modification, protein turnover, chaperones’ (10.15%), ‘signal transduction mechanisms’ (9.17%), ‘translation, ribosomal structure and biogenesis’ (6.10%), ‘energy production and conversion’ (5.88%), ‘carbohydrate transport and metabolism’ (5.77%), ‘transcription’ (5%), and ‘intracellular trafficking, secretion, and vesicular transport’(4.98%).

### 2.4. Comparative Analyses of Differentially Expressed Genes (DEGs)

DEGs were analyzed using the reads per kilo base per million (RPKM) method to determine the degree of overlap among the three sample groups. Venn diagrams were used to summarize the number of DEGs among the differential expression gene sets (DEGs sets), including A_B, A_C, and B_C. The total number of DEGs identified between group A and group B was 2641, including 1169 up-regulated and 1472 down-regulated genes, while the total number of DEGs between group A and group C was 3566, including 2160 up-regulated and 1406 down-regulated genes, and the total number of DEGs between group B and group C was 1098, including 851 up-regulated and 247 down-regulated genes (Figure 3a). Moreover, 159 DEGs (90 commonly up-regulated and 10 commonly down-regulated) were shared among the three DEG sets (Figure 3b–d).

### 2.5. Enriched KEGG Pathway Analyses of DEGs in Response to Low Temperature

KEGG is a signal pathway database for the systematic analyses of metabolic pathways and the functions of gene products in cells. The significance of enrichment was calculated by Fisher test, and enrichment of DEGs was analyzed at a significance level of *p*< 0.05 with an extremely rich signal pathway map. In the A_B gene set, 411 DEGs were assigned to the KEGG database involved in 110 pathways; in the A_C gene set, 627 DEGs were assigned to 117 pathways; and in the B_C gene set, 228 DEGs were assigned to 89 pathways. The 10 most reliable and significantly enriched pathways of DEGs in the three DEG sets are listed in Table 3. The enrichment degree of pathway was analyzed by enrichment factor. On integrating the numbers of differentially expressed genes in the A_B and A_C gene sets, seven major pathways related to cold acclimation mechanism were obtained (Additional File 2: Appendix A). They were ‘Photosynthesis-antenna proteins (KO 00196)’,‘Photosynthesis (KO 00195)’,‘Phenylalanine metabolism (KO 00360)’,‘Plant hormone signal transduction (KO 04075)’, ‘Galactose metabolism (KO 00052)’, ‘Phenylpropanoid biosynthesis (KO 00940)’, and ‘Starch and sucrose metabolism (KO 00500)’. As shown in Table 4, for the ‘Starch and sucrose metabolism’ and ‘Plant hormone signal transduction’ pathways, the number of DEGs in A_C was higher than in the A_B or B_C gene sets, but for the biological metabolism and synthesis pathways, there were no differences in the number of DEGs between the A_B and A_C gene sets.

### 2.6. TFs and PK Responding to Low Temperature

The gene expression network regulated by TFs plays an important role in the growth and development of plants. In this study, a total of 125 (64 up- and 61 down-regulated) and 184 (120 up- and 64 down-regulated) transcription factor genes (TFs) were identified in the A_B and A_C gene sets, respectively (Additional File 3: Appendix A). The key TF families involved in cold acclimation were AP2, Myb, WRKY, bHLH, Hlh, zinc finger, B3, LOB, bZIP, GRAS, and NAC. In gene set A_B, the Myb family was the largest group (16%), followed by the AP2 family (12%) and the NAC family (8.8%) (Figure 4a). In A_C gene set, the top three families were zinc finger (16.3%), Myb (13.04%), and AP2 (9.78%) (Figure 4b). The current study measured the differential expression of TFs by cluster analyses and identified 18 TFs that showed significant differences among different samples treated at different temperatures (Figure 4c, Additional File 3: Appendix A). The heat map of TFs showed that significantly up-regulated genes included two TFs members belonging to the NAC family (NAC2 and NAC29), two TFs-from the WRKY family (WRKY31and WRKY75), two TFs from the MYB family (MYB2 and MYB8), and two TFs from the zinc finger family (zinc finger2 and zinc finger3), identified in groups B and C (treated at 6 °C and low environmental temperature, respectively), while the significantly down-regulated TFs were mainly from the LOB and AP2 families (Figure 4c). 

A total of 90 and 117 DEGs encoding protein kinases were identified in the A_B and A_C gene sets, respectively. In the A_B gene set, there were 45 DEGs (22 up- and 23 down-regulated) encoding serine/threonine-protein kinase (STPK), including 10 DEGs of LRR receptor-like STPK, 9DEGs of CBL-interacting STPK, and 10 DEGs of G-type lectin S-receptor-like STPK. Besides, 4DEGs (3 up- and 1 down-regulated) belonging to the CBL-interacting protein kinase (CIPK) family, 7DEGs (all down-regulated) encoding heat shock protein (Hsp70 protein), and 4 DEGs (2 up- and 2 down-regulated) of receptor-like kinase (RLKs) were identified. In theA_C gene sets, we found 68 DEGs (40 up- and 28 down-regulated) encoding STPK, including 16 DEGs of LRR receptor-like STPK, 10 DEGs of CBL-interacting STPK and 12 DEGs of G-type lectin S-receptor-like STPK. In addition, 5 DEGs (all up-regulated) belonging to the CIPK family, 5 DEGs (all up-regulated) belonging to the mitogen-activated protein kinase (MAPKs) family, 6 DEGs (5 up- and 1 down-regulated) belonging to the mitogen-activated protein kinase kinase kinase (MPKKK) family, and 21 DEGs (3 up- and 18 down-regulated) of RLKs were also identified in the A_C gene sets. Details on these PK genes are provided in Additional File 4. 

### 2.7. Hormone-Related DEGs Responding to Low Temperature

A total of 26 DEGs in the A_B gene set and 40 DEGs in A_Cgene set were identified. These DEGs were involved in several plant hormone signal transduction pathways, including auxin, cytokinin (CK), gibberellin (GA), abscisic acid (ABA), and ethylene (ET), respectively. In the auxin signaling pathway, the transport inhibitor response 1 gene (*TIR1*) was down-regulated in the low temperature groups (B and C) compared with group A, butthe auxin-responsive protein gene (*IAA*) and the auxin-responsive GH3 gene (*GH3*) were down-regulated only in group B. In the CK pathway, the histidine-containing phosphotransfer protein gene (*AHP*) was down-regulated, and the two-component response regulator genes belonging to the ARR-B or ARR-A family (*ARR-B* or *ARR-A*) were up-regulated in the low-temperature groups (B and C) compared with group A. In the GA pathway, the gibberellin receptor GID1 gene (*GID1*) and the phytochrome-interacting factor 3 gene (*PIF3*) were up-regulated ingroups B and C. In the ABA signaling pathway, the ABA-responsive element binding factor gene (*ABF*), the sucrose non-fermenting 1-related protein kinase 2 gene (*SnRK2*), and the protein phosphatase 2C gene (*PP2C*) were up-regulated in group B, while only the *ABF* was up-regulated in group C (Figure 5a). The heat map of the hormone-related DEGs showed that the genes *T1R1*, *AHP*, *GH3*, and *IAA* were significantly down-regulated in the low-temperature groups (B and C) compared with group A, while the genes *GID1*, *SnRK2*, *PIF3*, *ARR-B*, *ARR-A*, and *PP2C* were significantly up-regulated in groups B and C (Figure 5b). Details of these genes above are provided in Additional File 5: Appendix A. The above results suggested that plant hormone signal transduction pathway played important roles in response to low-temperature stress in *C*. *oleifera* ‘Huaxin’.

### 2.8. Key DEGs Related to Photosynthesis

The DEGs were significantly enriched in two KEGG pathways, i.e., ‘Photosynthesis-antenna proteins’ and ‘Photosynthesis’ (Figure 6a,b), with the encoding proteins mainly located in the chloroplast. Cluster analyses revealed that the expression levels of all these DEGs significantly differed among the three groups (A, B, C).The genes encoding PSII oxygen-evolving enhancer protein 1 (psbO), PSII oxygen-evolving enhancer protein 3 (psbQ), PSII 10 kDa protein (psbR), PSII Psb27 protein (psb27), PSI subunit II (psaD), PSI subunit VI (psaH), PSI subunit PsaO (psaO), and F-type H+-transporting ATPase subunit delta (atpH) were more highly expressed in group A (25 °C), but were significantly decreased in the low temperature groups (B and C) (Figure 6c, Additional File 6: Appendix A). In addition, the expression levels of DEGs were consistent with the trends of photosynthetic physiological index such as Chl, *P_n_*, ETR, and the starch content (Figure 1d–1f, 1h), indicating that these DEGs might relate to photosynthesis. However, only the expression levels of PSII 22 kDa protein gene (*psbS*) were significantly increased in groups B and C compared with group A, indicating that *psbS* may play an different role during long-term acclimation. 

### 2.9. Key DEGs Related to the Membrane System and Osmoregulation

Low-temperature stress mainly damages the plasma membrane of plants. To sustain the functions of membrane proteins and lipids under cold stress, the plasma membrane needs to enhance lipid fluidity by increasing membrane lipids unsaturation levels [24], thus improving the cold tolerance of plants [25]. The current study identified five fatty acid desaturase (FAD) genes and three lipid-transfer protein (LTP) genes. Among them, four FADs were up-regulated in the group B (6 °C) and group C (low environmental temperature) while three LTPs were down-regulated compared to the group A (25 °C) (Additional File 7: Appendix A). Previous studies [24,25] have shown that soluble sugars, including maltose, trehalose, raffinose and glucose, were increased to maintain osmotic balance under cold temperatures, thus maintaining the stability of the osmotic regulation systems. In this study, the expression of several genes in related metabolism pathways also changed with temperature. Two trehalose-6-phosphate synthase genes (*TPSs*), two β-amylase genes (*AMYβs*), one α-amylase gene (*AMY*α), and one glucose-1-phosphate adenylyltransferase gene (*glgC*) involved in the starch and sucrose metabolism pathway, and two raffinose synthase genes (*RafS*) involved in the galactose metabolism pathway were all up-regulated in the low temperature groups (B and C) (Additional File 7: Appendix A). These are the key genes involved in sugar metabolism, all of which were expressed at the highest levels in group C but at the lowest levels in group A. However, one hexokinase gene (*HK*), three β-glucosidase genes (bgl*βs*) and two α-galactosidase genes (*gal*αs) related to glucose synthesis were down-regulated in groups B and C compared with group A. The expression of these genes, which are related to glycometabolism, indicates that they respond differently to low temperatures.

### 2.10. Other Important DEGs Responding to Low Temperature

KEGG pathway analyses revealed several enriched pathways associated with DEGs in response to low temperature. The current study identified other important DEGs related to cold stress, including one 4-hydroxyphenylpyruvate dioxygenase gene (*HPD*), two phenylalanine ammonia-lyase genes (*PALs*), and one cinnamate 4-hydroxylase gene (*CYP73A*) involved in phenylalanine metabolism pathway. In addition, three DEGs were identified involved in the flavonoid biosynthesis pathway, including flavonol synthase gene (*FLS*), flavonoid 3′-monooxygenase gene (*CYP75B1*), and chalcone synthase gene (*CHS*), with significantly different expression in the three sample groups. The expression levels of *FLS*, *CYP75B1*, *HPD*,and *CYP73A* were significantly higher in the low temperature groups (B and C) and lowest in the group A while the expression level of *CHS* was highest in the group A and lower in the low temperature groups (B and C) (Additional file 8: Appendix A). In addition, the galactinol synthase genes (*GOLSs*) involved in the galactose metabolism pathway were all up-regulated in groups B and C, while the gene of fatty acid hydroperoxide lyase (*HPL*) was significantly down-regulated. The expression of these genes changed in *C. oleifera* ‘Huaxin’ under low temperature, showing that these genes were induced by low-temperature stress.

### 2.11. Quantitative RT-PCR Analysis

To validate the reliability of the RNA-seq data, 12 target DEGs were selected and investigated by qRT-PCR. We compared the expression data of the 12 DEGs obtained by RNA-seqand qRT-PCR. The magnitude changes of these genes from three samples treated at different conditions observed by qRT-PCR were similar to that of RNA-seq (Figure 7), indicating that the RNA-seq data obtained in this study are reliable. However, differences in individual genes existed according to the RNA-seq and qRT-PCR results. For example, the expression levels of *amyB* (ID: c125403.graph_c2) and *psbS* (ID: c140493.graph_c0) were much higher in groups A and B when detected by qRT-PCR than as determined by RNA-seq, while the RPKM values of gene *Rafs2* (ID: c139678.graph_c1) appeared much lower in groups A and C when detected by qRT-PCR than as determined by RNA-seq. The differences may be due to the facts that some primer pairs used in qRT-PCR were not optimal for detecting the target transcripts.

## 3. Discussion

Tea-oil tree (*Camellia oleifera*) is an economically important trees pecies widely cultivated for edible oil production in China. *C. oleifera* is an evergreen tree species with a slow growth rate. It blossoms and yields fruits in the winter. Our previous study showed that low temperature had a great influence on the phenotype and physiology of *C. oleifera* ‘Huaxin’ leaves [19]. Normal flowering and fruiting require a lot of nutrients, which come from carbohydrate fixation by photosynthesis in the leaves. To better understand related genes expression of *C. oleifera* ‘Huaxin’ under low temperature stress, we performed transcriptomic analyses of leaves under long-term cold stress using Illumina NGS technology. The low growth rate of *C. Oleifera* ‘Huaxin’ results in only one main branch and no lateral branches at 2 years of age, while it can have 5–6 lateral branches at 4 years of age. Therefore, we chose 4-year-old potted young plants as they generate plenty of experimental materials.

Temperature is an important factor that restricts plant geographical distribution and influences plant growth performance. Unfortunately, most species in the genus *Camellia* are sensitive to low temperatures, which makes them difficult to overwinter in cold regions [26]. Exploring the molecular and physiological mechanisms of cold resistance can contribute to the development of cold-resistant *C*. *oleifera* varieties. The current study conducted a systematic transcriptome profiling of cultivated *C*. *oleifera* ‘Huaxin’ under simulated cold stress and low environmental temperature. Several DEGs involved in cold-responsive processes were identified, including cold-related transcription factor and protein kinase genes, hormone-related genes, photosynthesis-related genes, membrane system and osmoregulation-related genes, and other important enzyme-related genes. 

### 3.1. Transcription Factors and Protein Kinase Responding to Low Temperature

TFs are key factors involved in transmitting and amplifying low-temperature stress signals and regulating downstream resistance-related genes and their expression, thereby enhancing the cold tolerance of plants [27]. In this study, a large number of TF genes were involved in the low-temperature response. Generally, 11 families were identified with significant differential expression, including AP2, Myb, WRKY, bHLH, Hlh, zinc finger, B3, LOB, bZIP, GRAS, and NAC, most of which functioned as regulators related to cold stress responses [28]. The Myb and NAC families are generally considered the best-characterized classes of plant TFs, and many are involved in the cold response [29,30]. In this study, the Myb and NAC families were the two main groups that responded to cold stress, suggesting that *C*. *oleifera* ‘Huaxin’ might employ a complex mechanism of signaling and transcriptional reprogramming controlled by the Myb and NAC proteins. Moreover, the TF families were differentially expressed between simulated cold stress and low environmental temperature stress, which may have been due to the effects of other factors, such as light and moisture, or because the environmental temperature was much higher than the simulated temperature. 

Plant protein kinases belong to a large super family, and some play a central role in signal perception and transmission in response to cold stress. For example, CIPK is a key kinase involved in the Ca^2+^-signaling pathway in response to cold stimuli [31]. The CIPKs, MAPKs, and MPKKKs are involved in phosphorylation and de-phosphorylation processes, which have key roles in conferring cold tolerance by increasing the cold signaling activities [32]. Previous studies showed that RLKs are involved in the perception of environmental signals [33]. In this study, all of the CIPK, MAPK, and MPKKK genes were up-regulated under low temperature treatments compared to normal temperature treatment, confirming that they positively regulated resistance to cold stress, while most of the RLK genes were down-regulated, indicating that their protein products negatively regulated *C*. *oleifera* ‘Huaxin’. STPK enzyme functional activities are seen at the posttranslational level. Extensive STPK studies have been performed in pathogens, however, STPKs are implicated in the response to abiotic stress in higher plants [34,35]. Previously, Zorina [36] suggested that STPKs act as possible regulators of cold stress responses occurring at the transcriptional level in *Synechocystis*. In this study, genes mainly encoding three kinds of STPKs, including LRR receptor-like STPKs, CBL-interacting STPKs, and G-type lectin S-receptor-like STPKs were identified, and they were differentially expressed under low temperature stress; some were up-regulated and others were down-regulated compared to normal temperature. Because protein phosphorylation playsan important role in signal perception and transmission, these STPKs may participate in phosphorylation and de-phosphorylation processes that enable plants to survive in low-temperature environments. Therefore, these STPKs may act as cold stress sensors in the regulatory pathway of the cell response to cold stress. 

### 3.2. Hormone-Related DEGs Responding to Low Temperature

Phytohormones are small-molecule chemicals that are involved in the regulation of plant growth and abiotic stress responses [37]. Hormone-related DEGs in pathways such as CK, GA, ABA, and ET participate in the synthesis and metabolism of plant hormones, and these hormones may amplify existing cascades or initiate new signaling pathways [38]. In this study, four key response hormone signal transduction pathways were identified in cold acclimation of *C*. *oleifera* ‘Huaxin’ including CK, GA, ABA, and ET. The CK signaling pathway is a two-component system consisting of AHKs, AHPs, and ARRs. Type-A ARR genes are rapidly induced by CKs, and they regulate the activity of type-B ARRs via a negative feedback loop [39,40]. In this study, both *ARR-B* and *ARR-A* were up-regulated under low temperature stress, indicating that they might be key cold-responsive regulators that contribute to chilling tolerance. The ABA signaling pathway is a key regulator of the abiotic stress response in plants [41]. Low temperature can induce the expression of cold response genes (*CORs*), and ABA responds to cold stress by regulating these *CORs*. The expression of *CORs* depends on ABA-dependent and ABA-independent signaling pathways [28]. ABA-responsive element binding factor(ABF) can activate downstream gene expression in response to cold stress [42]. SnRK2 and PP2C are the main ABA receptors that form a signaling complex referred to as the “ABA signalosome”. The ABA signalosome can turn on ABA signals by phosphorylation of downstream transcription factors, such as the AREB/ABF or membrane proteins involved in ion channels [43]. Here, *ABF*, *SnRK2*,and *PP2C* were three key DEGs involving in ABA signal transduction pathway synthesis, all of which were up-regulated in response to low temperatures, suggesting that ABA biosynthesis and signaling components were important for the expression of cold-regulated genes(*COR*), and the expression of *CORs* in *C*. *oleifera* ‘Huaxin’ depends on ABA-dependent signaling pathway.

### 3.3. Key DEGs Related to Photosynthesis

The *Lhc* super-gene family is commonly divided into *Lhca* or *Lhcb*, depending on whether the gene products belong to the PSI or PSII super complexes, respectively. The *Lhca* super-gene family in green plants encodes several light-harvesting chlorophyll a/b-binding proteins, including Lhca1, Lhca2, Lhca3, Lhca4, and Lhca5, which harvest and transfer light energy to the reaction center of PSI in photosynthesis. The *Lhcb* super-gene family mainly encodes the heterotrimeric LHCII complex, including Lhcb1, Lhcb2, Lhcb3, Lhcb4, Lhcb5, Lhcb6, and Lhcb7, which are the most abundant membrane proteins on the Earth [44]. Low-temperature stress is necessary for photo-inhibition in chilling-sensitive plants such as pumpkin [45], maize [46], and sweetpepper [47]. In this study, photoinhibition was induced by long-term cold stress because Chl, ETR, and *P_n_* were significantly decreased under low temperatures compared to room temperature. Furthermore, the expression levels of *Lhca* and *Lhcb* genes significantly decreased under low temperatures, suggesting that low temperature may decrease the expression of *Lhca* and *Lhcb* decreased, thus resulting in the decrease of Chl, ETR, and *P_n_* and the starch content. PSII is a multi-subunit pigment-protein complex in higher plants and is mainly distributed in the stacking area of thylakoids. It consists of three key proteins including exogenous subunit II proteins PsbO (33 kD), PsbP (23 kD), and PsbQ (17 kD), which are encoded by genes of *PsbO*, *PsbP*, and *PsbQ*, respectively. PsbR is a small subunit protein in PSII that has important roles in polymerization stability, photosynthetic protection, and water oxidation of PsbP and PsbQ [48]. *PsbR* expression was inaccordance with the trends of ETR and *P_n_* in *C*. *oleifera* ‘Huaxin’ in respond to cold stress, indicating that *PsbR* may negatively regulate transcription during cold stress in *C*. *oleifera* ‘Huaxin’. This was further supported by previous studies showing that down-regulation of *PsbR* could lead to a decrease of *P_n_* and PSII photo energy conversion efficiency [49,50]. In addition, the reduced expression of *PsbR* could lead to reduced activity of the core coding proteins, such as PsbO, PsbP, and PsbQ in PSII, thus affecting the cooperation between PSI and PSII, and blocking the metabolic pathway of photosynthesis in leaves, resulting in decreased *P_n_* and ETR. Rorat [51] found that chilling treatment of potatoes induced a strongly enhanced accumulation of the *psbS* transcripts. Interestingly, in this study, only the expression levels of *psbS* were significantly increased under low-temperature stress, indicating that the expression pattern of the *psbS* gene was different from other genes related to photosynthesis, and it might have positive transcription regulation in response to cold stress in *C*. *oleifera* ‘Huaxin’. 

### 3.4. Key DEGs Related to the Membrane System and Osmoregulation

FADs and LTPs not only participate in the formation of biofilm, keratin, and lipids but also play important roles in regulating the unsaturation levels of membrane lipids and mediate the regulation of membrane fluidity in response to cold stress [52]. As important defense proteins in plants, the expressions of genes encoding FAD and LTP are induced by stress factors such as drought, high salinity, and low temperature. The plasma membrane is a primary site of injury from cold stress in plants. When plants are subjected to low temperatures, FADs and LTPs act as regulatory proteins to stabilize plasma membrane activity and protect it from cold injury [53]. In this study, genes encoding FADs and LTPs in *C*. *oleifera ‘*Huaxin’ were much more active under low environmental temperature than those under simulated low temperature, indicating that the genes were better induced by low environmental temperature, and too low temperature was not conducive to the expression of FADs and LTPs genes.

Maltose can function as a compatible solute, stabilizing substances in the chloroplast stroma in response to low-temperature stress, and there is a connection between the accumulation of maltose and the expression of the β-amylase gene (*AMYβ*), and low temperature can promote the hydrolysis of starch and increase the content of soluble sugar in cells [54]. Trehalose is a major osmotic protective substance present in organisms and some higher plants and can be accumulated in bacteria and yeasts in response to abiotic stresses [55]. Trehalose-6 monophosphate synthase (TPS) is the key enzyme synthesizing trehalose. In this study, the expressions of *AMYβ* and *TPS* were increased under low temperatures, which accelerated the degradation of the starch and promoted the synthesis of soluble sugar and galactose. These results were consistent with the altered starch content, soluble sugar content and galactose content, indicating that *C*. *oleifera* ‘Huaxin’ achieved its cold tolerance by accumulating maltose and trehalose at the flowering stage. Furthermore, the accumulation of the raffinose family of oligosaccharides (RFOs) enhances cold resistance in angiosperms. Besides, inositol galactose and raffinose could act as osmotic protective substances for plants under cold stress [56]. Inositol galactoside is the precursor of RFO synthesis, which is catalyzed by inositol galactose synthase (GOLS). Raffinose is synthesized by transfer of the galactosyl groups from inositol galactoside to sucrose and raffinose, respectively, which are catalyzed by raffinose synthase (RafS). Therefore, GOLS and RafS are considered key enzymes in the biosynthesis of RFOs. In this study, the expression of *GOLS* and *RafS* was up-regulated under low-temperature stress, suggesting that these genes are positively regulated at the transcription level in response to cold stress through the accumulation of osmotic protective substances such as inositol, galactose and raffinose. These results are consistent with previous studies, in which the expression of *GOLS* increased in vegetative tissues in relation to frost-tolerance and transportation of RFOs in *Ajugaretans* when exposed to cold stress [57]. In summary, solutes such as maltose, trehalose, raffinose, and inositol galactose accumulated through cold-inducible gene expression, thus maintaining the stability of the osmotic regulation systems.

### 3.5. Other Important DEGs Responding to Low Temperature

CYP73A5 and CYP75B1 are two types of monooxygenase enzymes belonging to the cytochrome P450 superfamily. Recently, increasing evidence has shown that P450s confer abiotic stress tolerance in plants [58,59,60]. *CYP73A5* and *CYP75B1* encode enzymes that are associated with the synthesis of phenylpropanoid in *Arabidopsis*. Furthermore, the CYP73A5 and CYP75B1 proteins mediate various steps in phenylpropanoid synthesis [58]. The current study identified two CYP genes (*CYP75B1* and *CYP73A*), which were up-regulated under low temperature stress. We assume that over expression of *CYP75B1* and *CYP73A* might contribute to the cold resistance of *C*. *oleifera* ‘Huaxin’ by mediating the synthesis of phenylpropanoid. Flavonoids are other plant products that have multiple biological functions, and their accumulation usually occurs in plants when they are subjected to abiotic stresses [61]. Anthocyanins and flavonols are two of the most important flavonoids that accumulate in response to low temperatures [62,63]. *CHS* and *FLS* are two key flavonoid biosynthetic genes, which encode key regulatory enzymes that participate in anthocyanin synthesis [64,65]. Increased gene expression of *PAL* and *CHS* might enhance cold tolerance in *Fagopyrum tataricum* and *C*. *sinensis* [22,66]. In addition, over accumulation of antioxidant flavonoids enhances ROS-scavenging activity under abiotic stresses [67,68]. In this study, the expression levels of *CHS* and *FLS* in *C*. *oleifera* ‘Huaxin’ were significantly increased under low temperatures, confirming that they enhanced cold tolerance in *C*. *oleifera* ‘Huaxin’ during cold acclimation.

## 4. Materials and Methods

### 4.1. Plant Materials and Low-Temperature Treatments

On February 18, 2016, 60 two-year-old *C. oleifera* cultivar ‘Huaxin’ (2009) young plants were selected and transplanted into plastic containers (22 × 22 × 20 cm) filled with a 2:1:1 mixture of peat soil, loess, and perlite. The plants were grown in field conditions with the same water and fertilizer management at the Life Science Building of Central South University of Forestry and Technology, Changsha, China (28°10′ N, 113°23′ E). Two years later, on November 3, 2018, 54 four-year-old *C. oleifera* ‘Huaxin’ with similar growth rates were divided randomly into three groups. Each group consisted of 18 plants. Those three groups of *C. oleifera* ‘Huaxin’ potted plants were placed in three different temperatures for the experiments: normal temperature of 25 °C in an artificial climate chamber as control (A); low temperature of 6 °C in an artificial climate chamber (B); and low environmental temperature in field conditions (C). Other parameters in each room of the artificial climate chamber A and B were the same with 70% relative humidity, a 12-h photoperiod at a photosynthetic photon flux density of 200 μmol∙m^−2^∙s^−1^, and an average CO_2_ concentration of 450 μmol∙mol^−1^ [19]. In the field condition (C), the average air temperature wasapproximately11.2 °C, the average relative humidity was 82.6%, and the average photosynthetically active radiation (PAR) was 360 μmol·m^–2^·s^–1^ on a cloudy day, and an average CO_2_ concentration of 392 μmol∙mol^−1^.

### 4.2. Determination of Physiological Indexes of C. oleifera ‘Huaxin’ under Low Temperatures

At 25 days after transplanting (25 DAT), the photosynthetic parameters, including Chl, *P_n_*, and ETR, were measured following Gu et al. [11] using a Li-6400xt (LI-COR Biosciences, Lincoln, NE, USA). The parameters of the portable photosynthesis system were set with a PPFD source provided by LEDs (Light Emitting Diodes) emitting red-blue light at 1000 μmol·m^–2^·s^-1^. Meanwhile, the CO_2_ concentration was 400 μmol·mol^−1^, and the flow rate was 500 µmol·s^–1^. The contents of soluble sugar and starch were measured as described by Wang et al. [69]. The content of galactose was determined using ELISA kits in accordance with the manufacturer’s instructions (Meimian industrial Co., Ltd., Jiangsu, China). The relative leaf water content (RWC) was determined according to the following equation: RWC= ((FM – DM)/(SM – DM))×100%. FM: leaves fresh mass; SM: leaves saturated mass; DM: leaves dry mass. The saturated mass was obtained after 24 h of leaf immersion in distilled water in the dark [70]. Nine seedlings for each treatment (three repetitions per treatment; three plants per replicate) and the leaves used for the measurements were kept under the same growth conditions and in similar growth positions. 

### 4.3. Sampling for RNA-Seq and RNA Preparation

During 25 DAT the plants underwent different temperatures, intact mature leaves were selected for the physiological experiments and molecular sequencing analyses. For each treatment, three samples were selected, which were named A1, A2, A3, B1, B2, B3, C1, C2, and C3, respectively. Then the total nine samples were rapidly frozen in liquid nitrogen and stored at −80 °C. A total of nine samples were used for the RNA-Seq and differential expression analyses. To avoid gene expression differences among leaves from the respective plants, the RNAs from three leaves belonging to the same plant were grouped for transcriptome sequencing. The purity, concentration, and integrity of the RNA samples were tested using an agarose gel and the Nanodrop 2500 (Thermo Fisher Scientific, Waltham, MA, USA) to ensure that the samples were suitable for transcriptome sequencing [71]. 

### 4.4. Library Preparation and RNA Sequencing

RNA samples were sent to Beijing Baimeike Biotechnology Co., Ltd. (Shunyi District, Beijing, China), where the libraries were produced and sequenced. For sample preparation, 1µg RNA per sample was used. Sequencing libraries were generated according to the instructions in the NEBNext^®^ Ultra™ RNA Library Prep Kit for Illumina^®^ (NEB, San Diego, CA, USA), and index codes were added to attribute sequences to each sample. Briefly, mRNA was purified from total RNA using poly T oligo-attached magnetic beads. Fragmentation was carried out using divalent cations under elevated temperature in NEB Next First Strand Synthesis Reaction Buffer (5X). First-strand cDNA was synthesized using random hexamer primers and M-MuLV Reverse Transcriptase. Second-strand cDNA synthesis was subsequently performed using DNA polymerase I and RNase H. Any remaining overhangs were converted into blunt ends using exonuclease/polymerase. After adenylation of the 3′ ends of the DNA fragments, the NEBNext Adaptor with a hairpin loop structure was ligated to prepare the DNA for hybridization. To select cDNA fragments of ~240 bp in length, the library fragments were purified with AMPure XP (Beckman Coulter, Beverly, MA, USA). Next, 3 µL of USER Enzyme (NEB) were used with size-selected, adaptor-ligated cDNA at 37 °C for 15 min, followed by 5 min at 95 °C before the PCR. Then PCR was performed with Phusion High-Fidelity DNA polymerase, Universal PCR primers, and Index (X) Primer. The PCR products were purified (AMPure XP), and the library quality was assessed using an Agilent Bioanalyzer 2100.

### 4.5. Transcriptome Assembly and Gene Functional Annotation

The left files (read1 files) from all libraries/samples were pooled into one big left.fq file and the right files (read2 files) were pooled into one big right.fq file. Transcriptome assembly was accomplished based on the left.fq and right.fq using Trinity [72] with min_kmer_cov set to 2 by default and all other parameters set to default. Gene function was annotated based on the following databases: NR, Pfam, KOG, COG, eggNOG, Swiss-Prot, KEGG, and GO.

### 4.6. Differential Expression Analyses 

Gene expression levels were estimated by RSEM [73] for each sample, in which fragments per kilobase of transcript per million mapped reads (FPKM) were used to estimate gene expression levels. Differential expression analyses of two sample groups at different temperature conditions were performed using the DESeq2 [74]. DESeq2 provides statistical routines for determining differential expression in digital gene expression data using a model based on the negative binomial distribution. The Benjamin Hochberg method was used to correct the significance *p*-value obtained from the original hypothesis test, and finally the corrected *p*-value (FDR, false discovery rate) was used as the key indicator of screening differential expression gene. In this study, the FDR< 0.01 and FC (fold change) ≥2 were set as the screening indexes for significantly differentially expressed genes. The differential expression gene set was obtained through analyzing the differential expression between two samples groups, and the gene sets were named A_ B, B_C and A_C in the analysis results.

### 4.7. Quantitative Real-Time PCR (qRT-PCR) Analysis

To validate the accuracy of the RNA-seq results, a total of 12 DEGs related to cold acclimation were selected for qRT-PCR validation, and the ACTIN11 (Glyma.18g290800) was used as the housekeeping gene. The total RNA was extracted from the same plant used for RNA sequencing. Three biological replicates were carried out for each treatment with three technological replications for each gene. The cDNA was synthesized from 500ng total RNA in a 20-µL reaction system by Aidlab reverse transcription Kit (TUREscript 1st Stand cDNA SYNTHESIS Kit). The specific primers were designed based on Primer Premier 5.0 (Additional file 9: Appendix A). Quantitative RT-PCR was performed using Analytikjena-Qtower 2.2 (Jena, Germany) in a final qRT-PCR reaction mixture containing 5.0 µL 2×SYBR^®^ Green Supermix, 1.0 uL primers, 1.0 µL cDNA, and 3.0 µL ddH_2_O. The calculation of the relative expression of the target gene in each sample was conducted using the 2^−ΔΔCt^ method as described previously [75]. The relative expression values of qPCR and RPKM values of RNA-seq were compared.

### 4.8. Statistical Analyses

All data were conducted with three replicates, and the results presented are the mean values. Microsoft Office Excel 2013 was used to process the data. The data of photosynthetic physiology were subjected to a one-way analysis of variance (ANOVA) with SPSS 17.0 software using Duncan’s multiple range test at the 0.05 level of significance. Other analyses of significance are illustrated in the corresponding section of the relevant method introduction.

## 5. Conclusions

The current study presents a transcriptome profile of cultivated *C*. *oleifera* ‘Huaxin’ using Illumina NGS technology to combine physiological data, which contributes to the understanding of the underlying transcriptional response mechanism of *C*. *oleifera* ‘Huaxin’ to low temperatures. Large numbers of DEGs in *C*. *oleifera* ‘Huaxin’ are induced by long-term cold stress, leading to phenotypic differences. The current study explored a series of potential chilling-resistant genes associated with photosynthesis, signal transduction, biofilm systems, and osmotic regulation systems, which can serve as candidate genes for cold tolerance in molecular breeding programs of *C*. *oleifera* ‘Huaxin’. Long-term cold stress could trigger the establishment of cold-resistance mechanisms and ultimately improve the adaptability of *C*. *oleifera* ‘Huaxin’ to cold stress.

## Figures and Tables

**Figure 1 ijms-21-00846-f001:**
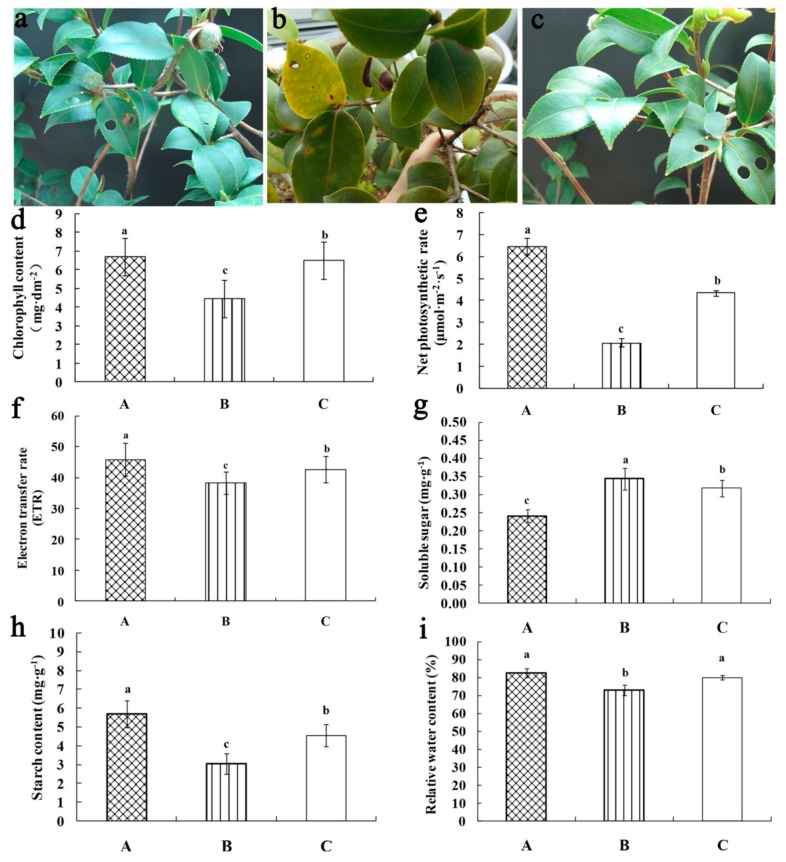
Effect of different temperaturetreatments on phenotypic changes, chlorophyll contents, net photosynthetic rate (*P_n_*), electron transfer rate (ETR), soluble sugar, starch content, and relative water contentin *Camelliaoleifera*‘Huanxin’ leaves. (**a**) 25 °C; (**b**) 6 °C; (**c**) Low environmental temperature; (**d**) Chlorophyll content; (**e**) Net photosynthetic rate; (**f**) Electron transfer rate; (**g**) Soluble sugar content; (**h**) Starch content; (**i**) Relative water content. Data represent the mean ± SE (*n* = 3). Lowercase letters indicate significant differences among different temperature treatments at *p* ≤ 0.05 according to Duncan’s multiple range test (DMRT).

**Figure 2 ijms-21-00846-f002:**
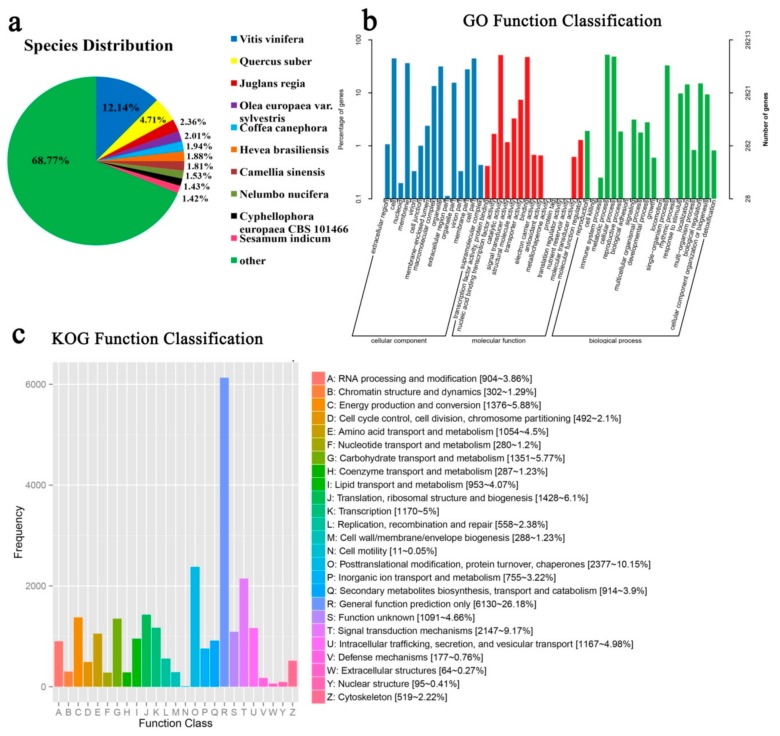
Functional annotations of the assembled transcriptome. (**a**) NR annotated species distribution map similar to the *Camellia oleifera* ‘Huaxin’ transcriptome; (**b**) GO classification of the annotated unigenes; (**c**) KOG function classification of the consensus sequence.

**Figure 3 ijms-21-00846-f003:**
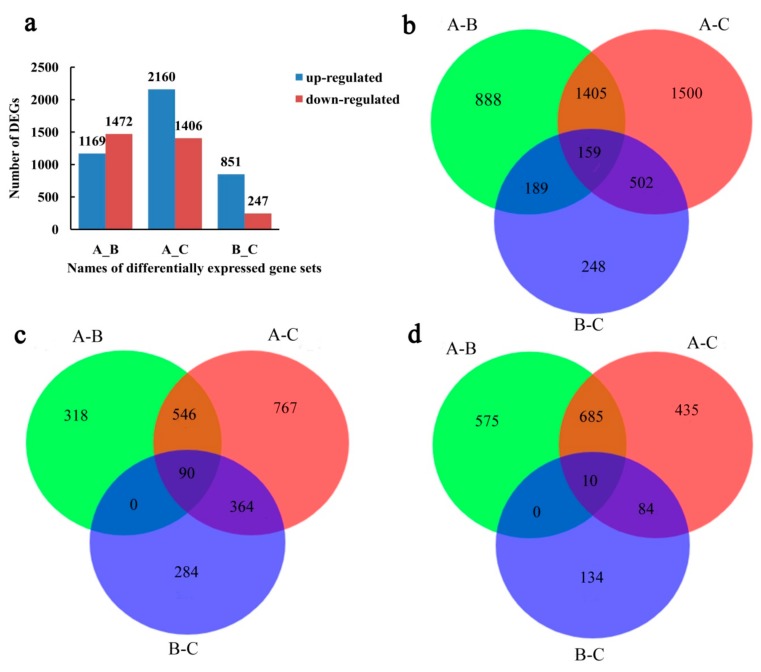
The graphical representation of differentially expressed genes (DEGs) of *Camellia oleifera* ‘Huaxin’ in response to low-temperature stress. (**a**) Number of up/down-regulated DEGs in the A_B, A_C, and B_C differential expression gene sets; (**b**) Venn diagram showing all DEGs numbers among the three gene sets; (**c**) Venn diagram showing up-regulated genes DEGs numbers among the three gene sets; (**d**) Venn diagram showing down-regulated DEGs numbers among the three gene sets.

**Figure 4 ijms-21-00846-f004:**
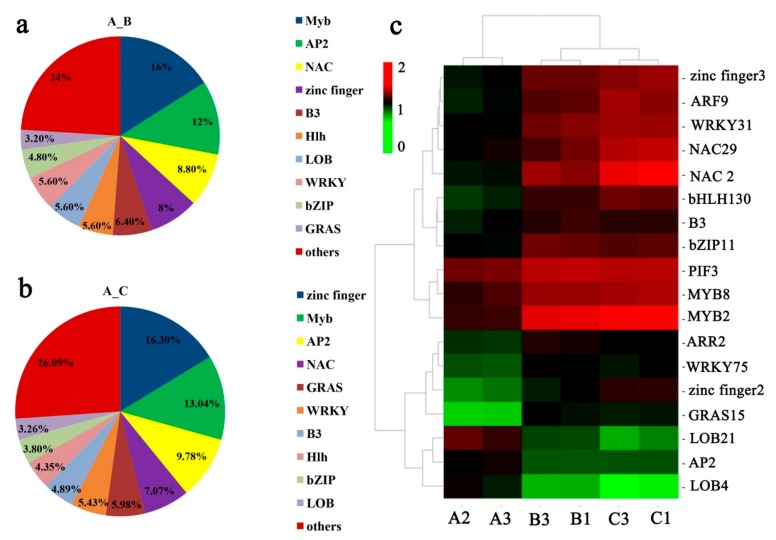
Analyses of the transcription factors (TFs) involved in *Camellia oleifera* ‘Huaxin’ response to low-temperature stress. (**a**) Distribution of transcription factor families in the A_B gene set; (**b**) Distribution of transcription factor families in the A_C gene set; (**c**) Expression profiles of 18 differentially expressed TFs among different samples. The heat map was generated from the log10 (FPKM values). Changes in expression level are represented by a change in color; green indicates a lower expression level, whereas red indicates a higher expression level. FPKM: fragments per kilobase of transcript per million mapped reads

**Figure 5 ijms-21-00846-f005:**
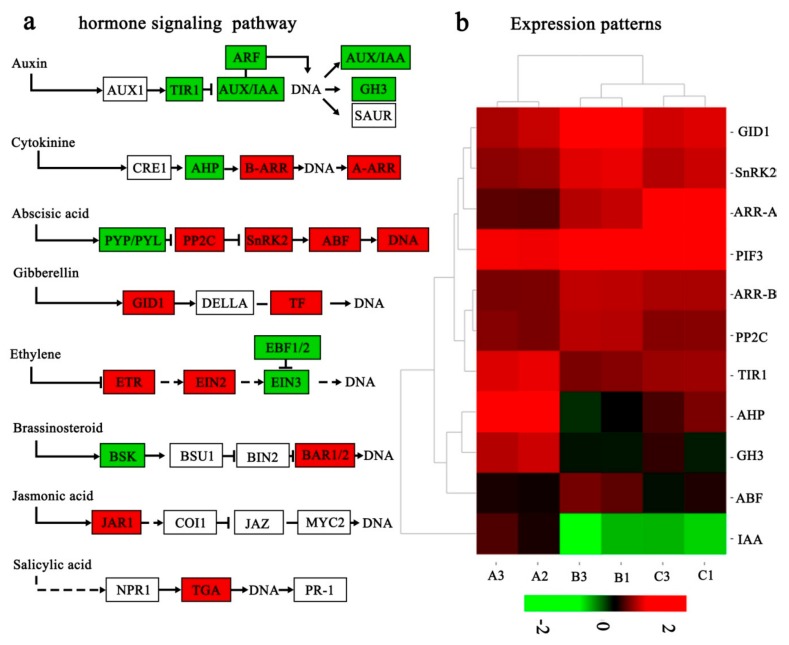
The differentially expressed genes in the hormone signaling pathways in response to low temperature in *Camellia oleifera* ‘Huaxin’. (**a**) The green and red boxes show the mapped differentially expressed genes in the plant hormone signaling pathways, as modified from a Kyoto Encyclopedia of Genes and Genomes map (KO 04075), and the green boxes indicate down-regulated genes, whereas red boxes indicate up-regulated genes. (**b**) The expression patterns are shown as a heat map. Changes in expression level are indicated by a change in color, green indicates a lower expression level, whereas red indicates a higher expression level.

**Figure 6 ijms-21-00846-f006:**
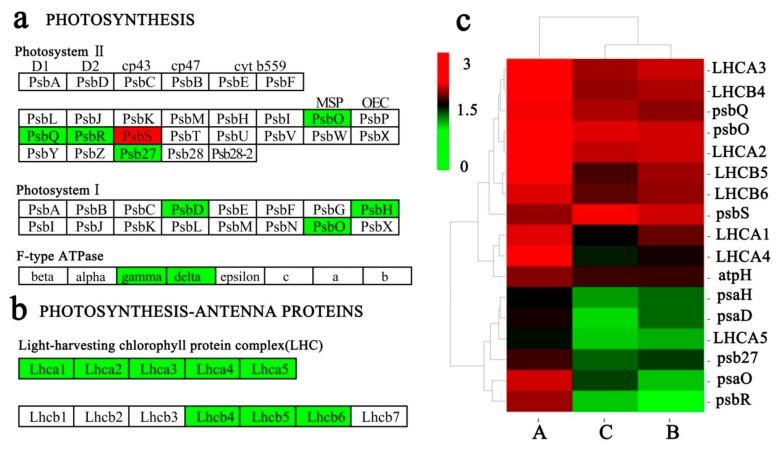
Differentially expressed genes (DEGs) related to photosynthesis pathways in response to low temperature in *Camellia oleifera* ‘Huaxin’. (**a**) and (**b**) Green and red boxes show the mapped DEGs in the ‘Photosynthesis’ and ‘Photosynthesis—antenna proteins’ pathways, as modified from the KEGG maps (KO 00195 and KO 00196). (**c**) A comparison of the gene expression patterns. Changes in expression level are indicated by a change in color, green indicates a lower expression level, whereas red indicates a higher expression level.

**Figure 7 ijms-21-00846-f007:**
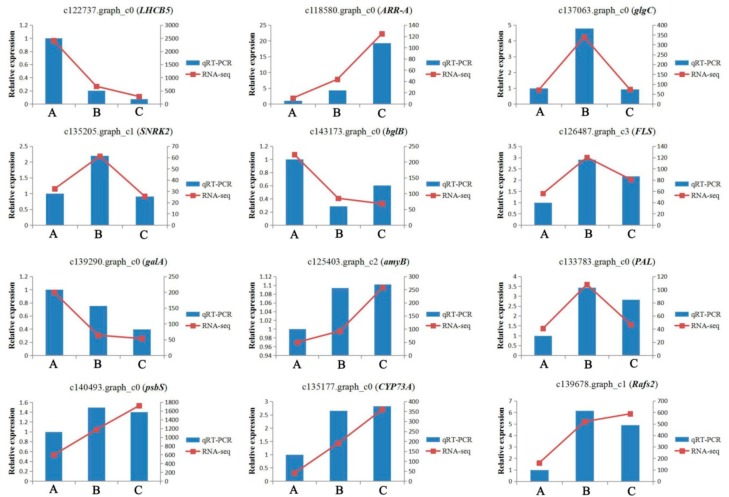
The comparison of the expression levels of 12 DEGs identified related to cold acclimation between RNA-Seq and qRT-PCR analyses from three samples treated at different temperatures.

**Table 1 ijms-21-00846-t001:** Length distribution of the transcripts and unigenes clustered from de novo assembly.

Length Range	Transcript	Unigene
200–300	42,879 (22.43%)	35,944 (35.69%)
300–500	35,226 (18.43%)	21,988 (21.83%)
500–1000	43,723 (22.87%)	18,930 (18.80%)
1000–2000	42,472 (22.22%)	14,590 (14.49%)
2000+	26,850 (14.05%)	9251 (9.19%)
Total number	191,150	100,703
Total length	197,720,185	79,598,970
N50 length	1690	1413
Mean length	1034.37	790.43

**Table 2 ijms-21-00846-t002:** Unigene function annotation statistics of *Camellia oleifera* ‘Huaxin’ transcriptome. COG: Clusters of Genes; GO: Gene Ontology; KEGG: Kyoto Encyclopedia of Genes and Genomes; Pfam: Protein family; KOG: EuKaryotic Orthologous Groups; NR: NCBI non-redundant protein sequences.

Annotated Databases	Annotated Number	Length <= 300 bp	Length >= 1000 bp
COG	13,161	3273	7010
GO	28,213	9019	12,485
KEGG	14,824	4606	7355
KOG	23,416	7249	11,409
Pfam	26,264	7364	14,330
Swiss-Prot	24,203	7277	13,150
eggNOG	40,098	13,082	17,864
NR	43,518	14,758	18,596
All	44,610	14,946	18,668

**Table 3 ijms-21-00846-t003:** The most reliable top ten significantly enriched pathways of DEGs in the three differential expression gene sets.

DEF-Sets	Pathway ID	Pathway	DEGs in Pathway	All Genesin Pathway	*p*-Value
A_B	ko00196	Photosynthesis—antenna proteins	10	30	9.95 × 10^−7^
	ko00941	Flavonoid biosynthesis	11	48	1.68 × 10^−5^
	ko00130	Ubiquinone and other terpenoid–quinone biosynthesis	11	59	0.000127
	ko00195	Photosynthesis	13	86	0.00029
	ko00360	Phenylalanine metabolism	12	76	0.000327
	ko00940	Phenylpropanoid biosynthesis	20	184	0.000769
	ko04075	Plant hormone signal transduction	26	271	0.000916
	ko00052	Galactose metabolism	13	107	0.002352
	ko00945	Stilbenoid, diarylheptanoid, and gingerol biosynthesis	5	20	0.002448
	ko00350	Tyrosine metabolism	10	72	0.002749
A_C	ko00196	Photosynthesis—antenna proteins	15	30	6.93 × 10^−10^
	ko00500	Starch and sucrose metabolism	44	281	2.45 × 10^−6^
	ko04075	Plant hormone signal transduction	40	271	2.95 × 10^−5^
	ko04070	Phosphatidylinositol signaling system	16	103	0.004376
	ko00940	Phenylpropanoid biosynthesis	24	184	0.005951
	ko00350	Tyrosine metabolism	12	72	0.007305
	ko00460	Cyanoamino acid metabolism	12	72	0.007305
	ko04712	Circadian rhythm—plant	12	74	0.00909
	ko00360	Phenylalanine metabolism	12	76	0.011198
	ko00960	Tropane, piperidine, and pyridine alkaloid biosynthesis	8	42	0.012192
B_C	ko03010	Ribosome	35	635	5.29 × 10^−5^
	ko00500	Starch and sucrose metabolism	19	281	0.000262
	ko00941	Flavonoid biosynthesis	7	48	0.000307
	ko00052	Galactose metabolism	9	107	0.002673
	ko00196	Photosynthesis—antenna proteins	4	30	0.008731
	ko00940	Phenylpropanoid biosynthesis	11	184	0.012611
	ko04626	Plant–pathogen interaction	14	262	0.01331
	ko00945	Stilbenoid, diarylheptanoid, and gingerol biosynthesis	3	20	0.016597
	ko00520	Amino sugar and nucleotide sugar metabolism	11	213	0.033104
	ko00905	Brassinosteroid biosynthesis	2	14	0.055219

**Table 4 ijms-21-00846-t004:** DEG statistics in seven major pathways of *Camellia oleifera* ‘Huaxin’.

Pathway	Pathway ID	A_B	A_C	B_C
Photosynthesis—antenna proteins	ko 00196	10	15	4
Photosynthesis	ko 00195	13	11	2
Phenylalanine metabolism	ko 00360	12	12	5
Plant hormone signal transduction	ko 04075	26	40	11
Galactose metabolism	ko 00052	13	13	9
Phenylpropanoid biosynthesis	ko00940	20	24	11
Starch and sucrose metabolism	ko 00500	23	44	19

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
