# Peer review of "Transcriptomic Analyses of *Camellia oleifera* ‘Huaxin’ Leaf Reveal Candidate Genes Related to Long-Term Cold Stress"

_ijms, 2020, doi:10.3390/ijms21030846_

Round 1

Reviewer 1 Report

Appreciate Author and team for good research work in Lab and as current research manuscript. But There are minor suggestions by the reviewer in the attached file. Hope an improved version of the manuscript.  

Author Response

Dear Reviewers:

Thank you for your comments concerning our manuscript entitled “Transcriptomic analyses of Camellia oleifera ‘Huaxin’ reveal candidate genes related to long-term cold stress during the flowering period” (ID: ijms-651006). The comments are valuable and helpful for revising our manuscript. We have studied the comments carefully and made appropriatecorrections. Revised parts are marked in red in our manuscript. The following is our response to the reviewers’ comments:

Point 1: Commented [A7]: Please rewrite the sentence and make clear the outcome of current study. Also the way forward for the future research in continuation of current research findings.

Response 1: We have significantly changed this part according to your suggestion. We have changed the text to “In summary, the results of DEGs analysis together with qRT-PCR tests contribute to our relevant researches on cold tolerance and further exploring new candidate genes for chilling-tolerance in molecular breeding programs of C. oleifera.”

Point 2: Commented [A8]: Please recheck as inference is missing ormis leading - woody oil or tea oil ????? or both It seems missing in current sentence – please recheck the reference

Response 2: We have changed this part according to your suggestion. We also added the reference at this section in the manuscript.

Point 3: Commented [A14]: Please make clear – strong fruiting meansno fruit fall ?? and resistance against whom??? Fruit falling or cold stress???

Response 3: We have changed this sentence. We have changed the text to “‘Huaxin’ is a new, high-yielding C. oleifera cultivar bred from common C. oleifera seedlings, which has high and stable yields, strong disease-resistant ability, and precocity.”

Point 4: Commented [A15]: Please add the valid reference (please avoid general statement)

Response 4:We have added valid reference in the manuscript.

Point 5: Commented [A18]: Please add clear research objectives of the current study.

Response 5:We have changed the text to “In this study, to explore the mechanism of C. oleifera leaves in response to long-term cold stress, a fully characterized C. oleifera transcriptome using Illumina NGS technology combining with physiological experiments were performed, and numbers of DEGs were identified and the gene expression patterns were compared between two temperature treatments and control, which contribute toabetter understanding of the molecular mechanism of low-temperature adaptation”

Point 6: Commented [A19]: These lines should be part of discussion of conclusion – it here in Introduction – please delete

Response 6: We have deleted the sentence in the manuscript according to your suggestion.

Point 7: Commented [A25]: Please shift this line to discussion part – this should not be part of result part.

Response 7: We have shifted this line to discussion part in the manuscript according to your suggestion.

Point 8: Commented [A28]: Please add the valid references of previous studies.

Response 8: We have added valid reference of the treatments in the manuscript.

Point 9: Commented [A38]: Please rewrite the sentence – please discussand relate with current research findings

Response 9: We havechanged this part according to your suggestion. We have changed the text to “In this study, genes encoding FADs and LTPs in C. oleifera were much more active under low environmental temperature than under simulated low temperature, indicating that these genes were better induced by the field conditions, and too low temperature was not conducive to the expression of FADs and LTPs genes.”

Point 10: Commented [A40]: Please add the parentage of Huaxin in bracket and also year of varietal release

Response 10: We have added the year of varietal release of Huaxin. In addition, Huaxin is a good, clonal grafting plant. Please see reference 15.

Point 11: Commented [A41]: Please add more details of these three treatments (A, B & C) – how these differentiate with each other

Response 11:We have added more details of these three treatments (A, B & C) as follows: Four year-old C. oleifera potted plants were placed in three different temperatures for the experiments: (A) normal temperature of 25°C in an artificial climate chamber, (B) low temperature of 6°C in an artificial climate chamber, (C) in field conditions.

Point 12: Commented [A42]: Please add a valid reference for these selected controlled environment condition (if any)

Response 12:We have added the valid reference, which is ourprevious research thathas been accepted by International Journal of Agriculture and Biology.

Point 13: Commented [A45]: Please valid reference of these treatment (ifany)

Response 13:We have added the valid reference in the manuscript.

Point 14: Commented [A46]: Please add the location name of thisinstitute

Response 14:We have added the location name of Beijing Baimeike Biotechnology Co., Ltd (Shunyi District, Beijing, China)

Point 15:Commented [A47]: Please valid reference of these treatment (ifany)

Response 15:We have added valid reference of these treatments in the manuscript.

Finally, we have corrected other errors related to grammar and spelling using the track changes mode in the manuscript according to the reviewers’ suggestions.

We greatly appreciated your help and hope therevised manuscript is acceptable for publication in your journal.

Sincerely,

Xiaofeng Tan, Ph.D.

Professor

Reviewer 2 Report

The authors described the investigation they conducted to assess the genes that were differentially expressed in the leaves of Camellia oleifera.

#1) The first issue is the authors attributed thr esponse to cold stress to genes differentially expressed in leaves. To be fair, the scope of the manuscript should be clearly set.  

#2) The authors did not clearly motivate their study; they particularly stressed the negative effects of low temperatures on the reproductive organs of Camellia oleifera, but they did not include the reproductive organs in their transcriptomic analyses and this should be discussed. This should be discussed particularly as it is well known that stress response is highly tissue specific.

#3) Graphics are of poor quality and their resolution should be improved.

Author Response

Dear Reviewers:

Thank you for your comments concerning our manuscript entitled “Transcriptomic analyses of Camellia oleifera ‘Huaxin’ reveal candidate genes related to long-term cold stress during the flowering period” (ID: ijms-651006). The comments are valuable and helpful for revising our manuscript. We have studied the comments carefully and made appropriatecorrections. Revised parts are marked in red in our manuscript. The following is our response to the reviewers’ comments:

Point 1: The first issue is the authors attributed the response to cold stress to genes differentially expressed in leaves. To be fair, the scope of the manuscript should be clearly set.

Response 1: We are appreciative of the reviewer’s suggestion. Since the results of DEGs analysis revealed a group of cold-responsive genes related to hormone regulation, photosynthesis, membrane systems, and osmoregulation, which contributes to our better understanding of the molecular mechanism of low-temperature adaptation and establish a basis for cultivar improvement and future molecular breeding of C. oleifera. Therefore, the scope of the manuscript belongs to  Molecular Plant Science or Molecular Biology.

Point 2: The authors did not clearly motivate their study; they particularly stressed the negative effects of low temperatures on the reproductive organs of Camellia oleifera, but they did not include the reproductive organs in their transcriptomic analyses and this should be discussed. This should be discussed particularly as it is well known that stress response is highly tissue specific.

Response 2: It is really true as the reviewer suggested that the reproductive organs should be discussed because of the negative effects of low temperatures on the reproductive organs of Camellia oleifera. Indeed, in our previous work [References 19: Effect of Chilling Temperature on Chlorophyll Florescence, Leaf Anatomical Structure, and Physiological and Biochemical Characteristics of Two Camellia oleifera Cultivars], which has been accepted by International Journal of Agriculture and Biology. We found that low temperature is not only conducive to the reproductive organs (especially the flowers), but also negatively affect the leaves of Camellia oleifera during the flowering period. In addition, the flower of C. oleifera Huaxin has withered and grown into young fruit at 25°C (Fig.1 a1),while a large number of Huaxin buds did not flower at 6°C, and remained in their original state (Fig.1 a3). Because the same flower organs grow and develop into different organs at different temperatures (Fig.1), we conducted transcriptomic analyses using the leaves of Camellia oleifera to reveal candidate genes related to long-term cold stress. Besides, the response of flowers to low temperature under a short period of time is what we're going to study next. We had a detailed discussion in the manuscript discussion.

Figure. 1: Please see the attachment word document.

Point 3Graphics are of poor quality and their resolution should be improved.

Response 3: As for the referee’s concern, graphics are of poor quality may due to being compressed in the Microsoft Word. Clear pictures were uploaded again, please see the attached file.

We have carefully revised the manuscript according to the reviewer’s suggestion by Dr. Zhiming Liu, a tenured Full Professor, Molecular Biologist, who has taught in the Department of Biology at Eastern New Mexico University, USA, for 20 years, has helped us in revising this manuscript.

Once again, thank you very much for allowing us to resubmit a revised version for publication in your prestigious journal.

Sincerely,

Xiaofeng Tan, Ph.D.

Professor

Reviewer 3 Report

The manuscript under review enumerates the long-term cold-induced transcriptome changes in Camellia oleifera ‘Huaxin’ var. during flowering.

The manuscript is reasonably well-written and explains the results obtained within its scope of discussion. Having said that, a few further informations would improve the manuscript significantly.

p15 l433:Authors discussed the upregulation of GolS/ RafS/ Tre genes from the transcriptome study and presented overall increase in soluble sugar under cold; which is a common response during cold-hardening. However, to clarify whether the above genes were responsible for the cold-hardening in this species, the determination of individual sugar content is important. The glucose/starch content also needs to determined, to correlate with reduced photosynthesis/ damaged photosynthetic apparatus. 

The title talks about cold stress during flowering period. The sample taken is from leaves. What is the significance of flowering period in this study and do the authors have data to support that the plant shows different transcript profile in other developmental stages? If not, I suggest to remove the phrase "flowering period" from the title. Also, if the data is taken during the flowering period, it would be noteworthy to document the changes in reproductive development when cold stress is applied.

The treatment B has visibly lost pigmentation after the cold stress. What is the osmotic potential of the plant at this stage ( can be determined by measuring RWC )? Are metabolic housekeeping genes functioning well (such as GAPDH ) in treatment B?

Author Response

Dear Reviewer 3:

Thank you for your comments concerning our manuscript entitled “Transcriptomic analyses of Camellia oleifera ‘Huaxin’ reveal candidate genes related to long-term cold stress during the flowering period” (ID: ijms-651006). Those comments are all valuable and very helpful for revising and improving our paper, as well as for providing important guidance to our research. We have made appropriate changes (use the track changes mode) to the manuscript according to your concerns. We hope that the corrections will satisfy your concerns. The following sare our responses to your comments:

Point 1: p15 l433:Authors discussed the upregulation of GolS/ RafS/ Tre genes from the transcriptome study and presented overall increase in soluble sugar under cold; which is a common response during cold-hardening. However, to clarify whether the above genes were responsible for the cold-hardening in this species, the determination of individual sugar content is important. The glucose/starch content also needs to determined, to correlate with reduced photosynthesis/ damaged photosynthetic apparatus. 

Response 1: We are appreciative of your suggestion. During the experiment, we carried out physiological and biochemical studies on the starch and galactose content. It is to be regretted that we didn't measure glucose content at that time for considering that the soluble sugar contains glucose, maltose, so we just determined the soluble sugar content. In this revised version, we have provided the data of starch content. The results showed that starch content of C. oleifera 'Huaxin' under treatment A was the highest among the treatments, and 85.67 and 24.73% higher than treatments B and C, respectively (Figure 1), the possible reason is that low temperature may lower the photosynthesis, and reduce the synthesis of starch. Furthermore, the expression level of β-amylase gene encoding amylase increased under low-temperature stress, which accelerated the degradation of the starch. We have made a relevant discussion in the manuscript.

Figure 1: Effect of different temperature treatments on starch content and relative water content in C. oleifera 'Huanxin' leaves. A: 25°C; B: 6°C; C: low environmental temperature. Data represent the mean ± SE (n = 3). Lowercase letters indicate significant differences among different temperature treatments at P ≤ 0.05 according to Duncan’s multiple range test (DMRT).

Point 2: The title talks about cold stress during flowering period. The sample taken is from leaves. What is the significance of flowering period in this study and do the authors have data to support that the plant shows different transcript profile in other developmental stages? If not, I suggest to remove the phrase "flowering period" from the title. Also, if the data is taken during the flowering period, it would be noteworthy to document the changes in reproductive development when cold stress is applied.

Response 2: It is true that we did not have data to support that the plant shows different transcript profile in other developmental stages. Hence, we have removed the phrase "flowering period" from the title. We have changed the title to “Transcriptomic analyses of Camellia oleifera ‘Huaxin’ leaf reveal candidate genes related to long-term cold stress”.

Point 3: The treatment B has visibly lost pigmentation after the cold stress. What is the osmotic potential of the plant at this stage ( can be determined by measuring RWC )? Are metabolic housekeeping genes functioning well (such as GAPDH ) in treatment B?

Response 3: It's a pity that we didn't measure the osmotic potential of the plant leaf at this stage, although I know that osmotic potential is closely related to plant water metabolism, growth and stress resistance. However, considering that the RWC is not suitable to correlate with the differential expression gene analysis, we did not present this index in our manuscript. Here we added the RWC in the first paragraph of the results section, and the result showed that relative water content (RWC) of treatment A was the highest. Compared to treatment A , RWC of treatment B was significantly decreased by 11.69%; however, no significant differences were found between treatments A and C (Figure 1). We also added the relevant discussion in the discussion section. Besides, the ACTIN11 was used as the housekeeping gene for qRT-PCR validation, and it functioned well in treatment B.

Once again, thank you very much for allowing us to resubmit a revised version for publication in your prestigious journal.

Sincerely,

Xiaofeng Tan, Ph.D.

Professor

Round 2

Reviewer 2 Report

I am presenting the review I completed on the manuscript titled “Transcriptomic analyses of Camellia oleifera ‘Huaxin’ reveal candidate genes related to long-term cold stress during the flowering period”.

With respect to the previous version the authors made some improvements but there is still more work to be done. Below are my comments:

OVERALL COMMENTS

#1. The research work performed is interesting but, the authors do not show they understand why they undertook the study in the first place. For instance, they are still not able to motivate why they chose to study the leaf transcriptome of Camelia oleifera to study cold response mechanisms of this species at the molecular level.

#2. A corollary to the above remark is that the title the authors gave to this manuscript is misleading. Instead, the true title should reflect clearly the fact that this work was based on the C. oleifera's leaf transcriptome. 

#3. English language including syntax should be revised. A particular effort was made only on the Abstract

#4. This manuscript presents a research work performed on C. oleifera, cultivar “Huaxin”. According to the authors’ account, this is “miracle” cultivar with superior performance under chilling conditions. The authors should briefly but, concisely, describe the history of this cultivar in the context of breeding so that breeders can know for instance, which breeding approaches were implemented to develop such cultivar.
A corollary to this is that in different parts of this manuscript, statements that strictly relative to the cultivar Huaxin are deliberately attributed to C. oleifera. This is like a pattern in the manuscript and should be avoided.

SPECIFIC COMMENTS

#5. Lines105-116: this statement is CLUMSY. Please consider replacing it with the following: "To explore the mechanism of C. oleifera leaf response to long term cold stress, a fully characterized C. oleifera transcriptome using Illumina NGS technology combined with physiological experiments was performed. This led to the discovery of several DEGs in response to cold and, ultimately, to a better understanding of the molecular mechanisms involved in the C. oleifera adaptation to low temperatures."

#6. Line 117: ... were identified to provide .... -> ... were identified that provide ....

#7. L 126: what do you mean by "and finally at A"; please re-formulate

#8. L 127-129: Please notice this is not English. please improve English language throughout the paper. I (and, the reader for that matter) should not be guessing your ideas.

#9. L132-136: Caption is poorly written, it should be reformulated. Abbreviations should be spelled out.

why do you use C. oleifera instead of the name of the variety? significant difference using which test?

#10. L 145-147: not classed but, classified. Why satisfactory? this should be measured against a previously accepted standard or one of your own.

#11. L 154: remove the redundant "unigenes"

#12. L 162: don't start a sentence with an abbreviation.

#13. L 167: (BP)) -> (BP); please read and double-check the manuscript for incorrections.

#14. L180: caption Table 2: please specify units

#15. L196: which genes? aren't there constitutively expressed genes? please get clear on this.

#16. Fig3a: this is not statistical analysis, it's a graphical representation/visualization. Is there any meaning tagged to the color code in the Venn diagram?

#17. L219: In Table 4 no statistical inferences were given. I noticed inferences in Table 3 are irrelevant for Table 4. Please provide (or otherwise discuss) the stat inferences for Table 4.

#18. L220: Please spell out abbreviated names the first time they appear in the manuscript.

#19. L237-240: Caption Fig 4: abbreviated names should be spelled out

#20. L277-279: Fig 5: same as above, please spell out abbreviated terms in the caption.

#21: L294-296: Fig 6: 1) I am not sure all these graphs are necessary in in this Figure. Most of them can be found in most textbooks. I strongly recommend to remove superfluous graphics/diagrams, and, as necessary, include relevant information in the text of the manuscript. 2) spell out the abbreviations.

#22. L312-317: How do you rate the information from say A_B, A_C, relative to the information from A, B, and C? This is a pattern throughout the manuscript and should be fixed. My understanding is that the transcriptomic information you present as A_B and A_C “batches” is not as informative as the information from A, B, and C groups. The point I want to make here is that we need to know which genetic factors are up/down-regulated under which conditions; otherwise it becomes confusing.

#23. L 352: what do you mean by "genetic breeding"? First time to hear. There are better expressions.

#24. L453-457: please discuss temperature levels not field conditions

#25. L498-506: this discussion is not satisfactory, and it is frankly, poorly written. First: you cannot discuss flower phenotypes because you did not present such data. second: The motivation of the tissue you selected for the transcriptomics should be well discussed in the first paragraph of the discussion section. This should look like a premise before further discussions. Please also discuss why you chose to conduct your study on 4-year old C. oleifera plants (refer to L508-520)

#26. L508-520: There are several confusions. what is the difference between plants used in 2016 and those in 2018? I can guess but you should clearly describe your experimental design. In addition: Field conditions is vague and general; please characterize field conditions in terms of temperature and the other environmental parameters recorded in the experimental rooms, so we can have a comparative idea relative to the other two media.

#27. Title should be revised as you are not validating qRT-PCR here. it's the other way around.

#28. L601-605: This section is oversimplified. There were more instances of statistical analyses that can be summarized here but taking care and avoiding repeating oneself when these methods were talked about in other sections. Precisely state where you applied which method. For instance Duncan test was used in one or very few instances in this work.

#29. L610: DEGsin -> DEGs in

Author Response

Dear Reviewer 2:

Thank you for your comments concerning our manuscript entitled “Transcriptomic analyses of Camellia oleifera ‘Huaxin’ reveal candidate genes related to long-term cold stress during the flowering period” (ID: ijms-651006). The comments are valuable and helpful for revising our manuscript. We have made appropriate changes (use the track changes mode) to the manuscript according to your concerns. The following sare our responses to your comments:

Point 1: #1. The research work performed is interesting but, the authors do not show they understand why they undertook the study in the first place. For instance, they are still not able to motivate why they chose to study the leaf transcriptome of Camelia oleifera to study cold response mechanisms of this species at the molecular level.

Response 1: Tea-oil tree (Camellia oleifera) is an economically important tree species widely cultivated for edible oil production in China. C. oleifera is an evergreen tree species with slow growth rate. It blossoms and yields fruits in the winter. Our previous study showed that low temperature had a great influence on the phenotype and physiology of C. oleifera ‘Huaxin’ leaves [19]. Normal flowering and fruiting need a lot of nutrients,which comes from carbohydrate fixation by photosynthesis in the leaves. To better understand related genes expression of C. oleifera 'Huaxin' under low temperature stress, we performed its transcriptomic analyses of leaves under long-term cold stress used Illumina NGS technology.  We have revised the section of introduction and discussion in the manuscript. (L395-404)

Point 2: #2. A corollary to the above remark is that the title the authors gave to this manuscript is misleading. Instead, the true title should reflect clearly the fact that this work was based on the C. oleifera's leaf transcriptome. 

Response 2: We have re-written the title according to your suggestion. We have changed the title to “Transcriptomic analyses of Camellia oleifera ‘Huaxin’ leaf reveal candidate genes related to long-term cold stress”.

Point 3: #3. English language including syntax should be revised. A particular effort was made only on the Abstract

Response 3: We have carefully revised the manuscript according to your suggestion by Dr. Heping Cao, a tenured full academic professor, plant physiologist and molecular biologist of the U.S. Department of Agriculture.

Point 4: #4. This manuscript presents a research work performed on C. oleifera, cultivar “Huaxin”. According to the authors’ account, this is “miracle” cultivar with superior performance under chilling conditions. The authors should briefly but, concisely, describe the history of this cultivar in the context of breeding so that breeders can know for instance, which breeding approaches were implemented to develop such cultivar.

A corollary to this is that in different parts of this manuscript, statements that strictly relative to the cultivar Huaxin are deliberately attributed to C. oleifera. This is like a pattern in the manuscript and should be avoided.

Response 4: We have added this part content according to the reviewer’s suggestion. Revised as follows:

In 1977, C. oleifera research team found a wild tea-oil tree with good fruit performance and strong resistance to disease in Chaling County, Hunan Province. From 1978 to 2009, through comparison test and regional test of 84 C. oleifera clones with good characters, the clones with the best comprehensive characters were screened out, which were named ‘Huaxin’ by the forest variety Committee of the State Forestry Administration. ‘Huaxin’ is a new C. oleifera cultivar, which has high and stable yields, strong disease-resistant ability, and precocity. In addition, we have changed 'C. oleifera' to 'C. oleifera Huaxin' in the manuscript according to the reviewer’s suggestion. (L75-81)

Point 5: #5. Lines105-116: this statement is CLUMSY. Please consider replacing it with the following: "To explore the mechanism of C. oleifera leaf response to long term cold stress, a fully characterized C. oleifera transcriptome using Illumina NGS technology combined with physiological experiments was performed. This led to the discovery of several DEGs in response to cold and, ultimately, to a better understanding of the molecular mechanisms involved in the C. oleifera adaptation to low temperatures."

Response 5: We have replaced this part according to the reviewer’s suggestion. (L104-109)

Point 6: #6. Line 117: ... were identified to provide .... -> ... were identified that provide ....

Response 6: We have changed “were identified to provide” to “were identified that provide”.

Point 7: #7. L 126: what do you mean by "and finally at A"; please re-formulate

Response 7: We have re-written this part according to the reviewer’s suggestion. We have changed the text to “all values were lowest for treatment B, followed by C, and the value of treatment A was the highest”(L120-122).

Point 8: #8. L 127-129: Please notice this is not English. please improve English language throughout the paper. I (and, the reader for that matter) should not be guessing your ideas.

Response 8: We have improved English language and corrected other problems of grammar and writing using the track changes mode in the manuscript according to the reviewer’s suggestion. We have changed the text to “However, the soluble sugar content of C. oleifera 'Huaxin' under treatments B and C were significantly higher compared with treatment A, which were 41.67 and 33.33% higher (Figure 1g).”(L122-126)

Point 9: #9. L132-136: Caption is poorly written, it should be reformulated. Abbreviations should be spelled out.

why do you use C. oleifera instead of the name of the variety? significant difference using which test?

Response 9: We are very sorry for our incorrect writing. We have changed the caption to “Figure 1. Effect of different temperature treatments on phenotypic changes, chlorophyll contents, net photosynthetic rate (Pn), electron transfer rate (ETR), soluble sugar, starch content and relative water content in C. oleifera 'Huanxin' leaves. A: 25°C; B: 6°C; C: low environmental temperature. Data represent the mean ± SE (n = 3). Lowercase letters indicate significant differences among different temperature treatments at P ≤ 0.05 according to Duncan’s multiple range test (DMRT).” (L132-138).

We also have changed “C. oleifera 'Huaxin'” instead of “C. oleifera” in the manuscript.

Point 10: #10. L 145-147: not classed but, classified. Why satisfactory? this should be measured against a previously accepted standard or one of your own.

Response 10: N50 is a common evaluation index for the integrity of de novo. It is defined as: after the reads are spliced, transcripts with different lengths will be obtained, then all transcripts are ordered by length from long to short. The total length of a transcriptome can be also obtained by adding all transcripts with different lengths. For example, ordered them as: unigene 25, unigene 24, unigene 23…… unigene 1, then add unigene in this order, when the added length reaches half of the total length of all unigene, the last unigene length added is N50 of the transcriptome(See the following figure for example). The longer N50 is, the better the assembly quality is.

Point 11: #11. L 154: remove the redundant "unigenes"

Response 11: We have removed the redundant "unigenes" in the manuscript.

Point 12: #12. L 162: don't start a sentence with an abbreviation.

Response 12: We have changed the text to “The GO is an international standardized classification system of gene functions”  in the manuscript.

 Point 13: #13. L 167: (BP)) -> (BP); please read and double-check the manuscript for incorrections.

Response 13: We are very sorry for our incorrect writing. We have revised all the incorrect writing in the manuscript.

Point 14: #14. L180: caption Table 2: please specify units

Response 14: We have added specify units in Table 2. (L184-185)

Point 15: #15. L196: which genes? aren't there constitutively expressed genes? please get clear on this.

Response 15: We have changed the text to "Moreover, 159 DEGs (90 commonly up regulated and 10 commonly down-regulated) were shared among the three DEG sets (Figure 3b–d), thus implying that these 159 DEGs might be responsible for C. oleifera 'Huaxin' responding to low temperatures."

Point 16: #16. Fig3a: this is not statistical analysis, it's a graphical representation/visualization. Is there any meaning tagged to the color code in the Venn diagram?

Response 16: We have changed the text to “The graphical representation of DEGs of C. oleifera 'Huaxin' in response to low-temperature stress. a, Number of up/down-regulated DEGs in the A_B, A_C, and B_C differential expression gene sets; b, Venn diagram showing all DEGs numbers among the three gene sets; c, Venn diagram showing up-regulated genes DEGs numbers among the three gene sets; d, Venn diagram showing down-regulated DEGs numbers among the three gene sets.”

The colour of green, red and blue refer to the differentially expressed genes (DEGs) in three gene sets as A_B, A_C, and B_C, while the other color codes mean the shared DEGs between two differentially expressed gene sets.

Point 17: #17. L219: In Table 4 no statistical inferences were given. I noticed inferences in Table 3 are irrelevant for Table 4. Please provide (or otherwise discuss) the stat inferences for Table 4.

Response 17: The enrichment degree of pathway was analyzed by enrichment factor, and the significance of enrichment was calculated by Fisher test (p< 0.05). Integrate the numbers of differentially expressed genes in A_B and A_C gene sets, seven major pathways related to cold acclimation mechanism were obtained. Please check the attachment for the enrichment results of KEGG pathway in Additional File 2: Figure S1.

Point 18: #18. L220: Please spell out abbreviated names the first time they appear in the manuscript.

Response 18: We have spelled out the abbreviated names according to the reviewer’s suggestion.

Point 19: #19. L237-240: Caption Fig 4: abbreviated names should be spelled out

Response 19: We have spelled out abbreviated names in the manuscript according to the reviewer’s suggestion.

Point 20: #20. L277-279: Fig 5: same as above, please spell out abbreviated terms in the caption.

Response 20: We have spelled out abbreviated names in the manuscript according to the reviewer’s suggestion.

Point 21: #21: L294-296: Fig 6: 1) I am not sure all these graphs are necessary in this Figure. Most of them can be found in most textbooks. I strongly recommend to remove superfluous graphics/diagrams, and, as necessary, include relevant information in the text of the manuscript. 2) spell out the abbreviations.

Response 21: We have removed the unnecessary figures in the manuscript.(L327)

Point 22: #22. L312-317: How do you rate the information from say A_B, A_C, relative to the information from A, B, and C? This is a pattern throughout the manuscript and should be fixed. My understanding is that the transcriptomic information you present as A_B and A_C “batches” is not as informative as the information from A, B, and C groups. The point I want to make here is that we need to know which genetic factors are up/down-regulated under which conditions; otherwise it becomes confusing.

Response 22: There were three different temperatures designed for this experiment: normal temperature of 25°C in an artificial climate chamber (A); low temperature of 6°C in an artificial climate chamber (B); in field conditions (C). Correspondingly, the nine samples used for RNA-seq were named A1, A2, A3, B1, B2, B3, and C1, C2, C3, respectively. There are numbers of differential expression genes (DEGs) between two different groups. We used A_ B, B_C and A_C to name the differential expression gene set in the DEGs analysis results. For example, A_ B referred to the differential expression gene set between A sample group (samples were treated at 25°C) and B sample group (samples were treated at 6°C).

Besides, according to your suggestion, we made changes include relevant information in the text of the manuscript, A_ B, B_C and A_C were used to illustrate the results of differentially expressed genes, and for other descriptions, we use A, B, and C sample groups in which samples were treated at corresponding temperature conditions to compare gene expression.

Point 23: #23. L 352: what do you mean by "genetic breeding"? First time to hear. There are better expressions.

Response 23: We have changed the text to “resist cold stress” instead of “genetic breeding” in the manuscript.

Point 24: #24. L453-457: please discuss temperature levels not field conditions

Response 24: We have changed the text to low environmental temperature in the manuscript.

Point 25: #25. L498-506: this discussion is not satisfactory, and it is frankly, poorly written. First: you cannot discuss flower phenotypes because you did not present such data. second: The motivation of the tissue you selected for the transcriptomics should be well discussed in the first paragraph of the discussion section. This should look like a premise before further discussions. Please also discuss why you chose to conduct your study on 4-year old C. oleifera plants (refer to L508-520)

Response 25: We have deleted this section, and we added discussion in the first paragraph of the discussion section. (L395-404)

Point 26: #26. L508-520: There are several confusions. what is the difference between plants used in 2016 and those in 2018? I can guess but you should clearly describe your experimental design. In addition: Field conditions is vague and general; please characterize field conditions in terms of temperature and the other environmental parameters recorded in the experimental rooms, so we can have a comparative idea relative to the other two media.

Response 26: We have rewritten this part according to your opinion.(L580-595)

Point 27: #27. Title should be revised as you are not validating qRT-PCR here. it's the other way around.

Response 27: We have changed the title to" Quantitative real-time PCR (qRT-PCR) analysis".

Point 28: #28. L601-605: This section is oversimplified. There were more instances of statistical analyses that can be summarized here but taking care and avoiding repeating oneself when these methods were talked about in other sections. Precisely state where you applied which method. For instance Duncan test was used in one or very few instances in this work.

Response 28: Here we mainly illustrate the data analysis methods not mentioned in the manuscript. For other significance analysis, we explained in the corresponding sections, for example, the significance analysis of differential expression gene used the Benjamin Hochberg method to correct the significance p-value, and finally used the adjusted p-value. Fisher's exact test was used to calculate the significance of enrichment in the analysis of differential expression gene in KEGG pathway.

We have changed the text to “All data were conducted with three replicates, and the results presented are the mean values. Microsoft Office Excel 2013 was used to process the data. The data related to photosynthetic physiology were subjected to a one-way analysis of variance (ANOVA) with SPSS 17.0 software using Duncan’s multiple range test at the 0.05 level of significance. For other significance analysis, we described in the corresponding method introduction.”

Point 29: #29. L610: DEGsin -> DEGs in

Response 29: We have changed the text to “DEGs in” instead of “DEGsin” in the manuscript.

Finally, we have corrected other errors related to grammar and spelling using the track changes mode in the manuscript according to the reviewers’ suggestions. We greatly appreciated your help and hope the revised manuscript is acceptable for publication in your journal.

Sincerely,

Xiaofeng Tan, Ph.D.

Professor

Reviewer 3 Report

Authors have mentioned in response 1 that they have already measured galactose/D-galactose content. In the light of the discussed genes, addition of that D-galactose content data would be informative for this line of work.

Author Response

Dear Reviewer 3:

Thank you for your comment concerning our manuscript entitled “Transcriptomic analyses of Camellia oleifera ‘Huaxin’ reveal candidate genes related to long-term cold stress during the flowering period” (ID: ijms-651006). Your comment is very helpful for revising and improving our paper, as well as for providing important guidance to our research. We have made appropriate change (use the track changes mode) to the manuscript according to your concern. We hope that the corrections will satisfy your concern. The following sare our response to your comment:

Point 1: Authors have mentioned in response 1 that they have already measured galactose/D-galactose content. In the light of the discussed genes, addition of that D-galactose content data would be informative for this line of work.

Response 1: We are appreciative of your suggestion. In this revised version, we have provided the data of galactose content. The results showed that the galactose content of C. oleifera 'Huaxin' under treatment B was the highest among the treatments, and 42.56 and 26.36% higher than treatments A and C, respectively (Figure S2). We added the figure of galactose in the manuscript, the figure is very large and difficult to typeset. Therefore, we will provide the figure of galactose as the attached figure in additional file 2-figure S1-2.

Once again, thank you very much for allowing us to resubmit a revised version for publication in your prestigious journal.

Sincerely,

Xiaofeng Tan, Ph.D.

Professor